# Insights on the Multifaceted Roles of Wild-Type and Mutated Superoxide Dismutase 1 in Amyotrophic Lateral Sclerosis Pathogenesis

**DOI:** 10.3390/antiox12091747

**Published:** 2023-09-10

**Authors:** Valentina Rubino, Giuliana La Rosa, Luca Pipicelli, Flavia Carriero, Simona Damiano, Mariarosaria Santillo, Giuseppe Terrazzano, Giuseppina Ruggiero, Paolo Mondola

**Affiliations:** 1Dipartimento di Scienze Mediche Traslazionali, Università di Napoli “Federico II”, Via Pansini 5, 80131 Naples, Italy; valentina.rubino@unina.it; 2Dipartimento di Medicina Clinica e Chirurgia, Università di Napoli “Federico II”, Via Pansini 5, 80131 Naples, Italy; giuliana.larosa@unina.it (G.L.R.); l.pipicelli@studenti.unina.it (L.P.); simona.damiano@unina.it (S.D.); marsanti@unina.it (M.S.); 3Dipartimento di Scienze, Università della Basilicata, Via dell’Ateneo Lucano 10, 85100 Potenza, Italy; flavia.carriero@unibas.it (F.C.); giuseppe.terrazzano@unibas.it (G.T.)

**Keywords:** Amyotrophic Lateral Sclerosis (ALS), Superoxide Dismutase 1 (SOD1), intracellular signaling, immune response regulation, excitotoxicity, neurodegeneration

## Abstract

Amyotrophic Lateral Sclerosis (ALS) is a progressive motor neurodegenerative disease. Cell damage in ALS is the result of many different, largely unknown, pathogenetic mechanisms. Astrocytes and microglial cells play a critical role also for their ability to enhance a deranged inflammatory response. Excitotoxicity, due to excessive glutamate levels and increased intracellular Ca^2+^ concentration, has also been proposed to play a key role in ALS pathogenesis/progression. Reactive Oxygen Species (ROS) behave as key second messengers for multiple receptor/ligand interactions. ROS-dependent regulatory networks are usually mediated by peroxides. Superoxide Dismutase 1 (SOD1) physiologically mediates intracellular peroxide generation. About 10% of ALS subjects show a familial disease associated with different gain-of-function SOD1 mutations. The occurrence of sporadic ALS, not clearly associated with SOD1 defects, has been also described. SOD1-dependent pathways have been involved in neuron functional network as well as in immune-response regulation. Both, neuron depolarization and antigen-dependent T-cell activation mediate SOD1 exocytosis, inducing increased interaction of the enzyme with a complex molecular network involved in the regulation of neuron functional activity and immune response. Here, alteration of SOD1-dependent pathways mediating increased intracellular Ca^2+^ levels, altered mitochondria functions and defective inflammatory process regulation have been proposed to be relevant for ALS pathogenesis/progression.

## 1. Introduction

Amyotrophic Lateral Sclerosis (ALS) is a progressive motor neuron degeneration associated with death of motoneurons of the spinal cord, brain stem and cortical pyramidal neurons network. This fatal neurodegenerative disease is not limited to α-motoneuron death since in many patients a cortical neuron degeneration and frontal dementia is present [1,2].

ALS is associated with muscle atrophy, weakness, fasciculations and spasticity [3] and sometimes accompanied by speech and swallowing dysfunctions leading to progressive paralysis and death due to respiratory failure.

Despite intense investigation efforts, little is known about the primum movens that generates ALS development.

More than 50 different genes have been linked to ALS [2,4,5,6,7]. Indeed, this neurodegenerative disease probably has a multifactorial genesis involving multiple mechanisms and transduction pathways, including oxidative distress [8,9].

As reported by Halliwell and Gutteridge, Reactive Oxygen Species (ROS) include both free radicals, with a very short half-life that ranges from 1 × 10^−6^ s to 1 × 10^−10^ s, and hydrogen peroxide (H_2_O_2_) [10], a relatively stable, non-radical derivative of oxygen molecule that is able to freely circulate inside cells and cross cell membranes. The ability of ROS to participate in the complex molecular network involved in the regulation of cell functional activity has been largely demonstrated in multiple cell/tissue contexts. Such regulatory functions have been observed to be essentially mediated by hydrogen peroxide, stable molecule that is able to freely diffuse inside cell membranes. SOD1, discovered by Mc Cord and Fridovich [11], is an ubiquitous CuZn binding dimeric enzyme, mainly localized in cytoplasm and also present in mitochondrial intermembrane space, involved in the physiological scavenging of O_2_^•−^ radicals, that are converted to molecular oxygen and hydrogen peroxide (H_2_O_2_) through the alternate reduction and re-oxidation of Cu^2+^.

SOD1 secretion has been observed in multiple cell models [12,13,14,15,16]; moreover, the ability of SOD1 to act as a signaling molecule, independent of its enzymatic activity, has been described [17,18].

Both redox stress and inflammation are associated with ALS pathogenesis/progression. Neuroinflammation represents a complex phenomenon, not only associated with the dysregulation of the glial cells surrounding neurons in the Central Nervous System (CNS), but also dependent on multiple regulatory molecular networks involving both innate and adaptive immune effectors [19,20,21]; furthermore, the role of T lymphocytes in inflammatory process regulation has been also demonstrated [22].

Hydrogen peroxide and superoxide anion have been described to participate in fine tuning of antigen-dependent immune response [23]. Indeed, activation of human T lymphocytes has been described to increase their SOD1 intracellular mRNA production and microvesicle secretion [23]. Moreover, in vitro administration of recombinant human SOD1 to activated human T cells has been observed to increase their production of the pro-inflammatory cytokine IL17 [24], playing a key role in the regulation of chronic inflammatory conditions [25,26].

Although ALS mostly affects patients without disease family histories, about 10% of ALS subjects show a familial form of this disease (fALS), mainly due to different gain-of-function mutations of SOD1 enzyme; indeed, the presence of a sporadic ALS (sALS), not clearly associated with SOD1 defects [27], has been described. However, in both conditions, the key events involved in ALS initiation and progression are still largely unclear.

Here, we propose that derangement of SOD1-dependent pathways, in neurons as well as in immune effectors, might play a key pathogenetic role in ALS and also in the absence of a mutated form of the enzyme.

## 2. Pathogenic Mechanisms of Mutant SOD1 in ALS

J.D. Atkin [28] demonstrated that mutated SOD1 (mSOD1) can inhibit secretory protein transport from the endoplasmic reticulum (ER) to Golgi apparatus, due to ER stress, Golgi fragmentation and altered axonal transport. The toxic effect carried out by SOD mutations in fALS has been attributed to its gain-of-function activity, in some cases associated with the occurrence of copper-dependent oxidative stress, largely demonstrated in patients affected by this condition. This “copper-dependent” oxidative hypothesis is challenged by the observation that the removal of copper chaperon from both wild-type SOD1 and mSOD1 is not effective in preventing motoneuron degeneration in transgenic mouse models of fALS [29]. In addition, it should be highlighted that SOD1 could itself be strongly oxidized, which may not only compromise the antioxidation function, but also acquire new binding and toxic properties [29,30].

Toxic effects of mSOD1 have also been related to its ability to produce insoluble intracellular aggregates, likely caused by the altered folding of the mutated protein, able to generate amyloid-like fibrillary structures, as proposed by the ‘aggregation hypothesis’ [31,32,33,34,35]. This hypothesis is particularly interesting because protein aggregation is frequently observed in ALS [36,37,38]. In addition, a zinc-deficient SOD1 has also been described to increase its aggregation ability upon oxidation [31], thus promoting the peroxynitrite-dependent protein tyrosine nitration, inducing motoneuron apoptosis [39]. Moreover, other studies, using transgenic rat/mouse models [40] as well as bone marrow transplants or chimeric animals, demonstrated that mSOD1 expression, in microglia cells, can also induce direct neuronal toxicity [41].

Between the several SOD1 mutations, the SOD1^G93A^ represents the mSOD1 molecule mostly used for in vitro and in vivo studies regarding human fALS [42,43]. In the transgenic SOD1^G93A^ ALS rodent model, unfolded protein response and ER stress-induced apoptosis have been observed [44].

Autophagy and mitophagy are conserved degradation mechanisms finalized to maintain cellular homeostasis and to respond to cellular stress conditions. These processes have been described to be specifically tuned in highly differentiated cells, like neurons [45]. Several genes implicated in control of autophagy and mitophagy have been observed to be associated with ALS. Thus, their involvement in the impairment of the protein clearance and in the formation of aggregates, severely damaging motor neurons of ALS patients, may be hypothesized [46]. Data on SOD1^G93A^ expressing neurons reveal in vitro a limited ability of such cells to respond to proteotoxic stress [47]. Moreover, in transgenic SOD1^G93A^ mouse models, the presence of a protein complex positive for p62, a product of the ALS-associated gene SQSTM1 that is linked with protein aggregates formation and to the presence of damaged mitochondria, has been observed [48].

## 3. ALS and Mitochondrial Damage

Cellular motoneuron damage in ALS has been described to be also associated with the mitochondrial alterations frequently observed in motor axon terminals of muscle biopsies of patients with early diagnosis of sporadic ALS. In fact, misfolded, metal-free SOD1 mutants form insoluble aggregates in motoneurons; such a condition represents an early event in the pathogenesis of ALS [49]. These mitochondrial alterations might generate an impairment in the respiratory chain metabolic processes, thus increasing oxidative stress induced damage [50,51].

Many SOD1 mutants show altered metal-binding ability; therefore, it has been suggested that metal-deficient SOD1, which is more readily taken up by mitochondria, may acquire copper in the intermembrane space causing protein aggregation and extensive mitochondria and cellular damages [52,53].

Physiological mitochondrial activity associates with outer and inner mitochondrial membrane separation, ensured by the integrity of the intermembrane space. Loss of such integrity generates a permeability alteration associated with the formation of the Mitochondrial Permeability Transition Pore (mPTP) [54,55]. Wild-type and mutant SOD1 have been observed to associate with mitochondria [47,56]. Post-translational oxidized wild-type SOD1 as well as its mutants have been observed to link Bcl-2, physiologically expressed in mitochondria, perturbing its structure and function with consequent release of cytochrome C and neuron death induction [30,57]. Moreover, this alteration generates an impairment of electron chain transport with further increased ROS production. This event might induce a pathological loop able to perturbate the redox mitochondrial homeostasis in motoneurons. SOD1^G93A^ has been observed, in mouse models, to access mitochondria matrix, thus inducing strong perturbation of oxidative phosphorylation [58,59,60,61].

## 4. Microglia Activation, NADPH Oxidase and SOD1 in ALS

Activated microglia represents a key element in Alzheimer’s disease as well as in the pathogenesis of ALS and other neurodegenerative conditions [62,63,64]. Microglia activation can be observed in transgenic mice expressing human SOD1 mutants, before neuron loss [65]; thus, dysregulated microglia functions, together with astrocyte activation, carry out an important role in ALS pathogenesis/progression [62,63,64,66].

ROS generated from NADPH oxidases play a role in signaling events leading to microglia activation [67]. Seven structural homologues of the phagocyte NOX enzyme (NOX2) have been identified, such as NOX1-5, DUOX1 and DUOX2 [68]. On the other hand, activated microglia produces ROS, primarily by NADPH oxidase 2 (NOX2), that result in enhanced microglia activation. It has also been shown that redox distress, caused by NOX1 and NOX2, significantly influences the progression of motor neuron disease, in mutant SOD1^G93A^ ALS mice [69]. Generation of ROS represents a general phenomenon in human cells. However, the excess of ROS production or an imbalance between ROS production and antioxidant defense are thought to represent important factors of disease progression in ALS [70,71].

Recent studies have shown that SOD1^G93A^ ALS transgenic mice generate high levels of gp91PHOX (NOX2) and superoxide in spinal cord microglia [72]. Moreover, it has been proposed that SOD1 interacts directly with Rac1, a cytosolic regulator of NOX2 [73], resulting in overproduction of ROS. In addition, ROS generated by Rac-dependent NADPH oxidases have been observed to be involved in cell signaling as well as in microbial killing.

Harraz [74] hypothesized that Rac-GTP-mediated activation of the NADPH oxidase complex might lead to production of O_2_^•−^ and H_2_O_2_, which were able to mediate Nox complex autoregulation by reducing Rac-GTP levels. Interestingly, Marden [69] showed that female ALS mice, lacking a copy of the X-chromosomal Nox1 or Nox2 genes, exhibited significantly increased survival rates, thus suggesting that a 50% reduction in NOX1/NOX2 expression levels might be associated with a substantial improvement of ALS outcome in mice. Moreover, the observation that multiple Nox genes might contribute to ALS progression clearly expands the potential therapeutic targets for this disease.

## 5. SOD1, Immunity and Neuroinflammation Processes in ALS

Chronic inflammation has been considered an important element in the pathogenesis/progression of different diseases, as well as in neurodegenerative processes. [75]. In this context, multiple immune dysfunctions, represented by extensive, dysregulated inflammatory processes, auto immunity phenomena, and deranged immune responses have been described in ALS. In addition, mutations in several genes, directly involved in immune response, have recently been reported in ALS patients [76,77,78,79].

Neuroinflammation, in particular, behaves as a key modulator of ALS progression potentially representing a prospective therapeutic target for this disease. In ALS, inflammatory responses are not restricted to the proximity of motoneurons but have been detected in muscles, peripheral axons, skin, liver and blood [80,81,82,83,84,85].

Multiple immune cell subsets have been described to participate in ALS pathogenetic mechanisms. Indeed, T cells, monocytes and other immune effectors have been observed to directly or indirectly access the CNS through the choroid plexus [86], thus mediating neuron damaging as well as neuroprotective processes.

E. Coque et al. [87] showed that ablation of CD8^+^ T cells in ALS mice increased the number of surviving motoneurons. Moreover, CD8^+^ T cells expressing the ALS-causing mSOD protein have been described to recognize and selectively kill motoneurons in vitro. To exert their cytotoxic function, CD8^+^ T cells carrying mSOD1 must be able to recognize neuron antigens inside MHC-I complex at the surface of the motoneurons. Moreover, analysis of the T cell receptor (TCR) diversity supports the evidence that self-reactive CD8^+^ T lymphocytes infiltrate the CNS of ALS mice and exert cytotoxic functions.

Fine-tuning of immune response is usually obtained by multiple regulatory processes, all belonging to the immune tolerance network, that are in place to prevent potentially deleterious immune responses against self-tissues. Regulatory T cells (Treg) are CD4 T lymphocytes characterized by the expression of the Foxp3 transcription factor. This T cell subset controls the immune effector response in terms of clonal expansion, differentiation, cytokine profile and tissue migration and is indispensable for the maintenance of immune self-tolerance [88]. Clinical studies in humans and in transgenic mouse models pointed out the role of Tregs in ALS pathophysiology. In humans and in mouse ALS models, Tregs infiltrating the Central Nervous System have been observed to suppress neuroinflammatory processes and promote the activation of neuroprotective microglia. Thus, immune-modulation strategies aimed at increasing Treg number and enhancing its functional effectiveness might be considered relevant to promote neuroprotective activity in ALS [89].

mTOR is an evolutionarily conserved serine/threonine protein kinase that directly influences T cell differentiation and proliferation by integrating environmental cues (nutrients, energy stores and growth factors) with immunity functions [90]. mTOR activity exerts opposite effects on effector T lymphocytes and on Treg. Indeed, mTOR inhibition strongly favors Treg differentiation and expansion while modulating T cell effector functions [91]. Compelling evidence indicate that progressive metabolic conversions, usually mediated by mTOR-dependent pathways, underlie the generation of proper effector functions during T cell response. Recent evidence indicates that SOD1 represents one of the major targets of the mTOR enzyme. Indeed, reversible mTOR-dependent SOD1 phosphorylation has been described to mediate SOD1 inhibition [91,92]. These observations propose a complex scenario in which SOD1/mTOR intracellular interplay may finely tune T cell activity [93,94]; instead, no data are available on the role of SOD1 in regulating mTOR dependent pathways. In this context, the recent correlation of increasing SOD1 intracellular levels in T cells with the presence of circulating Tregs in a cohort of subjects affected by Multiple Sclerosis undergoing effective immune-modulating treatment [24], strongly supports the idea that a SOD1-mTOR regulatory network [93,94] may participate in the complex mechanisms regulating immune-modulation processes.

Hydrogen peroxide and superoxide anion are involved in TCR-dependent signaling and adaptive immune response activation [23]. Indeed, antigen-dependent stimulation of human T lymphocytes can modify SOD1 intracellular localization in T cells, mediating a clear SOD1 recruitment by TCR clusters. In addition, increased SOD1 mRNA production and microvesicle secretion of the enzyme, by activated human T lymphocytes, has been observed [23].

We found [24] that in vitro administration of recombinant human SOD1 to activated human T cells increases their IL17 production, a key cytokine in induction/maintenance of chronic inflammatory processes [25,26]. This effect is mediated by SOD1-dependent enzymatic activity, since SOD1 molecule lacking dismutase activity (Apo SOD1), it is unable to affect T cell cytokine production. Furthermore, hydrogen peroxide addition to activated T-cells mimics the SOD1 effect [24]. These data indicate, as summarized in Figure 1, that SOD1 effects on inflammatory process may be, at least partially, linked to its hydrogen peroxide production; this molecule is more stable than other ROS and can freely migrate outside cells and between different cell compartments [24].

Thus, a very complex scenario involving multiple molecular networks, including SOD1-dependent pathways, might be involved in the regulation of T cell activation and differentiation in the neuroinflammatory context.

## 6. Constitutive and Inducible SOD1 Secretion

The first demonstration of constitutive SOD1 secretion in many eukaryotic cells date back to many years ago when we, for the first time, showed the export of this protein in hepatocytes and fibroblasts [95], in neuroblastoma SK-N-BE cells [96] and in thymus-derived epithelial cells [12]. Many others researchers confirmed our observations [13,14,15,16]. It is of relevance that although more than 33% of all proteins are exported through the ER and Golgi compartments [97], wild-type SOD1 is secreted by an alternative pathway, bypassing the canonical ER–Golgi apparatus [13].

SOD1 constitutively produced and released by microglia cells, by a lysosomal secretory pathway, has been also described to play a neuroprotective role [15]. Moreover, an association between impaired constitutive extracellular secretion of mutant SOD1, the presence of cytoplasmic insoluble protein inclusions and toxicity in NSC-34 cell line [98] has also been reported.

In the ALS transgenic rat model, the chronic intraspinal infusion of exogenous unmutated human SOD1 significantly delayed disease progression, suggesting a novel extracellular role for SOD1 in ALS pathogenesis and therapy [16]. In this context, a chromogranin-mediated secretion of mutant SOD1, but not wild-type SOD1 proteins, linked to ALS progression, has been observed [16], while a clear association of microglia activation with the occurrence of motor neurons toxicity has been revealed in transgenic mice carrying SOD1 mutations [16,99].

The constitutive SOD1 secretion suggests a paracrine antioxidant and protective role of this protein against oxygen radical in extracellular milieu. In addition, the mechanisms underlying the observed proinflammatory effects of SOD1 secretion [24] and the ability of the enzyme to potentiate the kinase activity in certain cell types by directly or indirectly modulating phosphatase functional effectiveness [100,101,102] need to be further investigated.

In vitro experiments performed by our group demonstrated that SOD1 is able to directly interact with cell surface of human neuroblastoma SK-N-BE cells activating a phospholipase C-dependent (PLC-PKC) pathway, with consequent massive intracellular Ca^2+^ increase; interestingly, this effect is independent from the dismutase activity of SOD1 molecule, since apo SOD1 (free metal SOD1) or mimetic SOD1 (MnTMPyP) were observed to mediate the same effects. Moreover, U73122, a powerful PLC inhibitor, was observed to strongly reduce SOD1-induced intracellular Ca^2+^ increase [17].

In addition to the constitutive SOD1 secretion, we showed that SOD1 is present in large dense core vesicles actively released from rat brain synaptosomes, as well as from rat pituitary GH3 cells. This rapid SOD1 export, mediated by depolarization and induced by high extracellular K^+^ concentration, is strictly associated with an increase in intracellular Ca^2+^-mediated SOD1 exocytosis; this SOD1 export has been observed to rely on the SNARE protein-dependent synaptic exocytosis machinery [103].

## 7. Role of SOD1 Interaction with the Muscarinic M1 Receptor

Accumulation of Ca^2+^ in neurons, with consequent activation of Ca-dependent pathways, has been largely associated with cell death [104]. Intracellular Ca^2+^ increase might be mediated by different mechanisms, as represented by several voltage-gate-dependent calcium channels (VGCCS), metabotropic (M1 muscarinic) receptors and ionotropic glutamate-dependent receptors (N-Methyl-D-aspartate (NMDA) and non-NMDA molecules). In addition, the intracellular Ca^2+^ increase can be mediated by the involvement of metabotropic glutamate receptors (GRM) [105]. Reduced glutamate re-uptake, mediated by altered transporter availability, has been also associated with increased Ca^2+^ concentration at neuron post-synaptic level. Moreover, Ca^2+^ overload inside mitochondria has been observed to induce deranged generation of ROS with subsequent availability of pro-apoptotic factors. Such altered ROS production may be involved in damaging processes along with ‘free radical buffering depletion’, mediated by glutamate competing with cysteine at the glutamate–cysteine pump [106].

Excitotoxicity has been defined as a phenomenon dependent on massive glutamate release related to depolarization, as well as on impaired glutamate re-uptake. Both mechanisms lead to over-activation of NMDA receptors thus inducing heavy intracellular Ca^2+^ influx [107]. Still largely unknown are the potential downstream effects due to interactions between mSOD1, M1 receptors and glutamate excitotoxicity.

Increased levels of glutamate in the cerebrospinal fluid, likely dependent on the selective decrease in the glial glutamate transporter EAAT2, associated with reduced glutamate transport into astrocytes, have been characterized in subjects affected by sporadic ALS [108]. In this context, excitotoxicity, generated by excessive glutamate levels, has been proposed as a relevant mechanism involved in ALS pathogenesis.

Our group recently found that SOD1 interaction with muscarinic M1 receptor, in human neuroblastoma SK-N-BE cells, can activate ERK1/2 and AKT kinases in a dose/time-dependent manner. This effect was strongly reduced by M1 receptor silencing, as well as by using the M1 antagonist pirenzepine. Moreover, the incubation of SK-N-BE and the neuroblastoma–spinal motoneuron fusion NSC-34 cell line with mutant SOD1^G93A^ significantly increased their intracellular Ca^2+^ concentration, as compared with wild-type SOD1 treatment [109].

Primary myocytes and skeletal muscle fibers derived from SOD1^G93A^ transgenic mice have been observed to show perturbed expression of Ca^2+^ transporters, likely responsible for their altered mitochondrial Ca^2+^ fluxes [110]. These defects occur in young SOD^G93A^ mice prior to the disease onset. Thus, mSOD1 mutants might be involved in the alterations in the skeletal myocytes’ functions largely reported in ALS.

The elevation of mitochondrial Ca^2+^ concentration, induced by muscle contraction or muscle inactivity, has been hypothesized to strongly interfere with ROS production. Moreover, temporal profile of mitochondrial Ca^2+^ intracellular levels has been suggested to behave as a physiological switch flipping between the beneficiary versus destructive outcomes [111,112].

The potential ability of SOD1 mutants to affect Ca^2+^ mitochondrial levels, consequently interfering with mitochondrial ROS production, needs to be investigated.

These observations, as a whole, are conceivable with the hypothesis, summarized in Figure 2, that the Ca^2+^-dependent excitotoxicity, associated with depolarization-dependent increase in SOD1 extracellular levels with consequent M1 receptor/SOD1 interaction, may be involved in the derangement of key neuro-modulatory networks during ALS pathogenesis. 

Five muscarinic receptor subtypes (m1–m5) have been identified in T lymphocytes [113,114]; m1, m3 and m5 muscarinic receptor subtypes are coupled to Gq/11, which, upon stimulation, mediate activation of phospholipase C activity, resulting in increases in Ca^2+^ availability inside cells [115]. Moreover, acetylcholine is synthesized and released by T-lymphocytes, acting as an autocrine/paracrine factor, likely involved in immune function regulation [115].

These observations add further complexity to the biological scenario involved in the regulation of the neuroinflammatory context.

## 8. Discussion

ALS is a progressive motor neurodegenerative disease whose pathogenetic mechanisms are still largely unknown. Gain-of-function mutations of SOD1 enzyme, physiologically involved in the intracellular generation of hydrogen peroxide, have been associated with the familial form of ALS, usually accounting for 10% of all the cases, involving multiple gene mutations.

Neuron functional effectiveness and survival have been largely described to depend on a complex molecular network involving intracellular neuronal and extra-neuronal circuits. The function of neuron-surrounding microglia cells as well as the presence of neuroprotective immune-dependent pathways, usually regulated by multiple cell subsets, have been extensively studied. In this context, a deranged regulation of each component of this complex scenario may be hypothesized to be relevant for ALS pathogenesis/progression.

Compelling evidence indicate that in familial and sporadic ALS, SOD1 misfolded proteins are able to mediate endoplasmic reticulum (ER) stress and fragmentation of the Golgi apparatus, resulting in impaired protein secretion and cellular apoptosis. Moreover, extensive mitochondrial alterations, with severe impairment in the respiratory chain metabolic processes, have been described to increase oxidative stress-induced damage [49,50,51].

SOD1 mutants have been largely demonstrated to maintain their enzymatic activity, suggesting that the role of mSOD1 in ALS is characterized by a gain-of-toxic function, rather than by a loss of function feature [116,117].

The secretion of SOD1 by normal and transformed cells has been previously reported [12,13,95,96]; in addition, our in vitro experiments demonstrated that SOD1, in SK-NB-E cells, activates the PLC-PKC pathway increasing intracellular Ca^2+^ concentration; these data are confirmed by the ability of the PLC inhibitor U73122 to revert SOD1-dependent Ca^2+^ increase in SK-NB-E cells [17].

The mechanistic difference between the pathogenesis of familiar and sporadic forms of ALS has not yet been clarified, since in the latter, misfolded wild-type SOD1 protein seems to activate the same neurotoxic pathway, that is mediated by SOD1 mutants in familiar ALS [118,119]. The mechanism by which mutated SOD1^G93A^ induces ER stress, Golgi apparatus fragmentation and accumulation of misfolded or unfolded proteins at the endoplasmic reticulum lumen has been extensively studied, but is still not quite clear [98,120,121].

In this review, we highlight the observations that SOD1 enzyme acts as a signaling molecule modulating different subcellular processes to adjust molecular homeostasis of the cell. SOD1 deficiency is associated with oxidative distress in Multiple Sclerosis, an autoimmune demyelinating disease, evidencing the importance of this molecule in maintaining normal function of the central nerve system [122].

Recent data obtained in SOD^G93A^ mouse ALS model [123] reveal that the treatment outcome may significantly vary with the disease progression. Indeed, targeting of oxidative stress in pre-onset, of excitability in near onset, of inflammation in post-onset, and of apoptosis in the end-stage condition have been shown to be associated with disease improvement. In this context, SOD1 molecule can be considered as a potential unifying molecular pathway, whose proper targeting might improve ALS treatment at different disease stages.

Many effects of SOD1 could be ascribed to its peroxide generation obtained by radical oxygen dismutation; in this context, hydrogen peroxide concentration shift, from physiological (in eustress condition) to pathological (higher H_2_O_2_) levels, may participate in SOD1-mediated pathological effects. In fact, in eustress condition, H_2_O_2_ can be considered as a key signaling molecule that is able to regulate many physiological functions in relation to its concentration, like innate and adaptive immunity activities [124], biosynthesis of thyroid hormones, multiple cells signaling pathways, gene expression and cellular growth [125,126].

Besides the role of H_2_O_2_ as a physiological signal transduction molecule, our in vitro experiments indicate that the availability of SOD1-containing extracellular microvesicles, dependent on the depolarization of excitable cells, induces a markedly higher extracellular SOD1 concentration, as compared with the SOD1 concentration released in the constitutive secretion. This effect generates an increase in intracellular Ca^2+^ concentration that is strongly enhanced by incubating excitable cells with SOD1^G93A^ [109].

Therefore, it can be conceivable that an enhanced export of SOD1^G93A^ from excitable cells, with consequent increased interaction with muscarinic M1 membrane neuron receptor, results in a significant increase in intracellular Ca^2+^ level associated with neuron cells apoptotic death, as suggested by our in vitro observations.

Thus, besides the Ca^2+^ cytotoxic effects induced by ROS and NMDA glutamate receptors stimulations, a further marked increase in intracellular Ca^2+^ concentration may be likely carried out by the deranged SOD1^G93A^ muscarinic M1 receptor interaction. Further experiments by depolarization of human neuroblastoma NS-K-BE cells transfected with SOD1^G93A^ are necessary to strengthen such hypothesis.

Neuron depolarization and antigen-dependent T cell activation both contribute to the exocytosis of SOD1, leading to an increased interaction of the enzyme with a complex receptor network. This interaction could play a crucial role in neuroinflammation during ALS pathogenesis.

The involvement of deranged inflammatory processes in motoneuron degeneration as well as the key role of immune-mediated neuroprotective pathways, closely related to T cell activity and differentiation, has been largely recognized. Indeed, T cell response in CNS has been described as able to mediate both neuroprotection as well as damaging effects on neuronal tissues [127,128,129]. In this context, Treg subset, physiologically involved in the tolerance control, has been observed to play a relevant role for neuroprotection against neuroinflammatory processes and for neurological recovery [128,130]. Our recent observations [24] have also been proposing the relevant role of SOD1 intracellular levels in Treg differentiation. Consistently, the ability of mTOR enzyme, a key regulator of Treg subset, to control SOD1 intracellular activity has been revealed [93,94].

## 9. Conclusions

The collective findings here support the notion that SOD1, including its gain-of-function mutant SOD 1 isoforms, plays a pivotal role in multiple cellular pathways, whose derangement might represent a key element in ALS pathogenesis/progression.

Specifically, derangement of SOD1-dependent pathways has been proposed to perturbate the complex immune regulatory mechanisms, likely crucial for the control of neuroinflammatory processes and neurological recovery. Moreover, excitotoxicity, largely described in ALS, may be enhanced by the ability of SOD1 and even more by its mSOD1^G93A^ mutant to generate, also by interaction with muscarinic M1 receptor, a pathological increase in intracellular Ca^2+^ levels, with consequent neuron apoptosis.

These observations bring new insights on the complex interplay between SOD1, neurotoxicity and neuroinflammation, with the potential to inspire novel therapeutic strategies to control the detrimental effects of SOD1 dysregulation in neurodegenerative disorders.

## Figures and Tables

**Figure 1 antioxidants-12-01747-f001:**
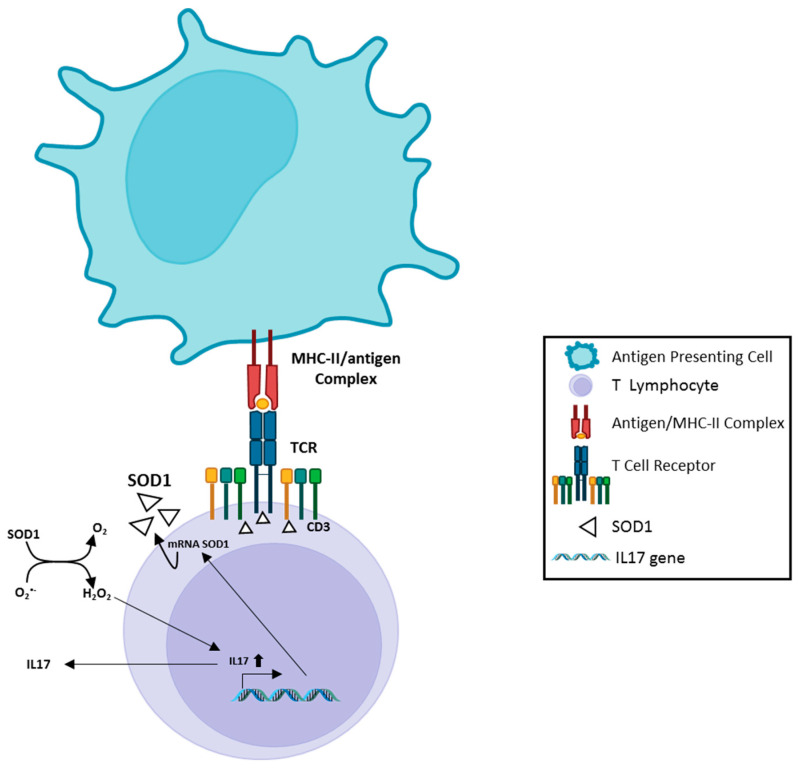
A scheme of SOD1 involvement in antigen-dependent T cell response. Recognition of antigen by TCR, inside MHC complex expressed on the surface of the Antigen Presenting Cell, induces rapid recruitment of SOD1 molecules near the TCR clusters; this phenomenon is accompanied by increased SOD1-mRNA, followed by SOD1 export by a microvesicle pathway. Extracellular SOD1 has been observed to mediate enhanced IL17 production by activated T lymphocytes. This effect depends on enzymatic activity of SOD1; the involvement of IL17 in induction/maintenance of chronic proinflammatory processes has been extensively studied by others. The image has been partially created by using BioRender.com.

**Figure 2 antioxidants-12-01747-f002:**
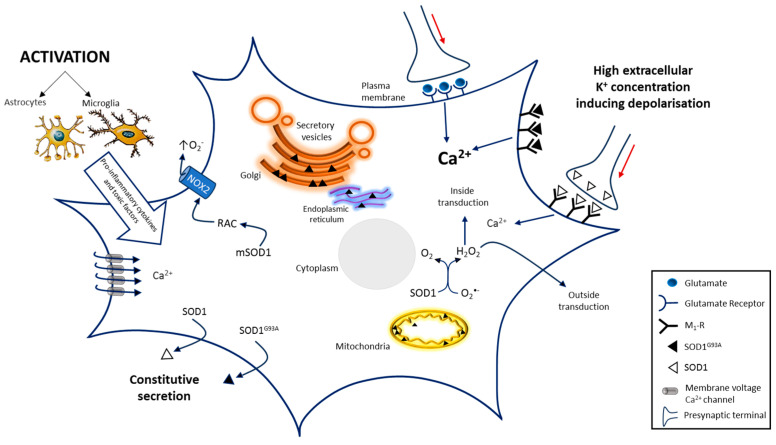
A scheme of the SOD1-dependent molecular mechanisms involved in motoneuron death in ALS. SOD1-dependent pathways may damage motoneurons increasing ROS production through RAC activation, that impairs protein export, by protein aggregation in the ER–Golgi apparatus and by inducing deranged mitochondrial activity generating further ROS production Excitotoxicity induced by glutamatergic synapses hyper-stimulation or by impaired glutamate transporter into glial cells and presynaptic membrane, also contributes to motoneuron death. Similar effects could be due to SOD1 and even more due to SOD1^G93A^ interaction with metabotropic muscarinic M1 receptor. Both mechanisms generate excitotoxicity by increasing intracellular Ca^2+^ concentration that alters multiple Ca^2+^-dependent signaling pathways. Activation of glial cells, generating pro-inflammatory cytokines and toxic factors, also contributes to the progression of ALS. Others factors involved in the pathogenesis of ALS progression are not represented. The red arrows indicate signal propagation inside axon; the increasing black indicate higher concentration. Image has been partially created by using BioRender.com.

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
