# Peer review of "Insights on the Multifaceted Roles of Wild-Type and Mutated Superoxide Dismutase 1 in Amyotrophic Lateral Sclerosis Pathogenesis"

_antioxidants, 2023, doi:10.3390/antiox12091747_

Round 1
Reviewer 1 Report

It is recommended to extensively rewrite the manuscript by an author with English as his/her first language.
Author Response
The authors certainly selected a very intriguing perspective of ALS pathogenesis for this review. Yet upsettingly many important mechanistic details were not explored in depth, merely scratching the surface of the phenomena
We thank the reviewer for his/her carefully evaluation of our manuscript.
ALS pathogenesis is characterized by a very complex scenario involving different, very heterogeneous, apparently unrelated molecular networks.
The aim of our analysis, focusing the physiological role of SOD1 and the potential mechanisms underlying the deranged activity of the SOD1 gain-of-function mutants, is to open new insights on the complex interplay between SOD1, neurotoxicity and neuroinflammation, in order to unveil novel therapeutic strategies to control the detrimental effects of SOD1 dysregulation in neurodegenerative disorders.
The revised paper, according with the reviewer’s suggestions, that we really appreciate, has been modified in order to more deeply discuss some critical issues, as represented by the SOD1 involvement in:
- the regulation of the autophagy/mitophagy processes
- the mitochondrial-dependent ROS production
- the immune tolerance control (mTOR/SOD1 regulatory pathways)
- the complex molecular network underlying the physiological control of the cytosolic calcium concentration in neurons in baseline and in depolarization-dependent conditions
All the reviewer’s remarks have been carefully considered. The revised manuscript has been modified, accordingly
More discussions are needed about:
- mSOD1 and autophagy/mitophagy abnormalities in motor neurons and other cell types;
Autophagy and mitophagy are conserved degradation mechanisms finalized to maintain cellular homeostasis and to respond to cellular stress conditions. Multiple molecular pathways are involved in the control of such key processes usually avoiding pathological accumulation/aggregation of damaged molecules. Growing evidences indicate that autophagy/mitophagy processes might be specifically tuned in highly differentiated cells, like neurons (Chantell S. et al; 2016). Several genes, implicated in autophagy and mitophagy control have been observed to be associated with ALS. Thus, their involvement in the impairment of the protein clearance and in the formation of aggregates, severely damaging motor neurons of ALS patients, might be hypothesized (Taylor JP et al; 2016). Data on SOD1G93A expressing neurons reveal in vitro a limited ability of such cells to respond to proteotoxic stress (Maday S. et al; 2012). Moreover, in transgenic SOD1G93A mouse models the presence of a protein complex positive for p62, a product of the ALS-associated gene SQSTM1, linked with protein aggregates formation and to the presence of damaged mitochondria, has been observed (Rudnick ND, et al; 2017).
As requested, the revised manuscript has been modified to include such data.
- How mSOD1 elevates mitochondrial ROS production and its association with mitochondrial permeability transition pore;
Several morphological and ultrastructural changes have been described in mitochondria of ALS subjects. However, the mechanisms underlying such alterations are still largely undefined. Physiological mitochondrial activity associates with outer and inner mitochondrial membrane separation, ensured by the integrity of the intermembrane space. Loose of such integrity generates a permeability transition associated with the formation of the Mitochondrial Permeability Transition Pore (mPTP) (Lee J. Martin et al 2009; Lee J. Martin, 2010). Wild type and mutant SOD1 have been observed to associate with mitochondria (Okado-Matsumoto, et al. 2001; A., Fridovich, I. et al , 2001; Higgins, C.M.J. et al 2002). Post-translational oxidized wild type SOD1 as well as its mutants have been observed (Ezzi SA, et al 2007; Guareschi S, et al 2012) to link Bcl-2, physiologically expressed in mitochondria, perturbing its structure and function with consequent release of cytochrome C and neuron death induction. Moreover, this alteration generates an impairment of electron chain transport with furtherly increased ROS production. This event might induce a pathological loop able to perturbate the redox mitochondrial homeostasis in motoneurons. SOD1G93A has been observed, in mouse models, to access mitochondria matrix, thus inducing strong perturbation of oxidative phosphorylation. Outer mitochondrial membrane extension and leakage with intermembrane space expansion, together with the formation of megamitochondria in motoneuron cell bodies have been described (Lee J. Martin et al 2009; Lee J. Martin, 2010).
As requested, the revised manuscript has been modified to include such data.
- How the inhibition of mTOR signalling induce Treg like gene expression profile and the expansion of Treg pool.
Fine-tuning of immune response is usually obtained by multiple regulatory processes, all belonging to the immune tolerance network, that are in place to prevent potentially deleterious immune responses against self-tissues. Regulatory T cells (Treg), are CD4 T lymphocytes characterized by the expression of the Foxp3 transcription factor. This T cell subset controls the immune-effector response in terms of clonal expansion, differentiation, cytokine profile, tissue migration and is indispensable for the maintenance of immune self-tolerance (Sakaguchi S, et al, 2006). mTOR is an evolutionarily conserved serine/threonine protein kinase that directly influences T cell differentiation and proliferation by integrating environmental cues (nutrients, energy stores and growth factors) with immunity functions (De Candia P et al 2022). mTOR activity exerts opposite effects on effector T lymphocytes and on Treg. Indeed, mTOR inhibition strongly favours Treg differentiation and expansion while modulating T cell effector functions. Compelling evidence indicate that progressive metabolic conversions, usually mediated by mTOR-dependent pathways, underlie the generation of the proper effector functions during T cell response (Procaccini C et al 2012; de Candia et al, 2022). Recent evidence indicate that SOD-1 represents one of the major targets of the mTOR enzyme. Indeed, reversible mTOR-dependent SOD-1 phosphorylation has been described to mediate SOD-1 inhibition (Tsang, C.K et al 2018). These observations propose a complex scenario in which SOD-1/mTOR intracellular interplay might finely tune cell activity. No data are available on the role of SOD-1/mTOR dependent pathways in immune tolerance control.
As requested, the revised manuscript has been modified to more deeply address such issue.
- The downstream effect of elevated cytosolic calcium due to interactions between mSOD1 and M1 receptors and its distinctions with glutamate excitotoxicity.
Accumulation of Ca2+ in neurons, with consequent activation of Ca-dependent pathways, has been largely associated with cell death (Floyd CL, et al. 2005). Intracellular calcium increase might be mediated by different mechanisms as represented by several voltage-gate-dependent calcium channels (VGCCS), the metabotropic (M1 muscarinic receptors) as well as by the ionotropic glutamate-dependent receptors (N-Methyl-D-aspartate (NMDA) and non-NMDA molecules). In addition, the intracellular calcium increase can be mediated by the involvement of metabotropic glutamate receptors (GRM). (Xiao B, et al 2019). Moreover, Ca2+ overload inside mitochondria has been observed to induce deranged generation of ROS with subsequent availability of pro-apoptotic factors. Such altered ROS production might be involved in damaging processes also through the ‘free radical buffering depletion’, mediated by glutamate competing with cysteine at the glutamate-cysteine pump (Lewerenz J, et al 2013). Excitotoxicity has been defined as a phenomenon dependent on massive glutamate release related to depolarization, as well as on impaired glutamate re-uptake. Both mechanisms lead to over-activation of NMDA receptors thus inducing heavy intracellular calcium influx (Luo T, et al 2011). Still largely unknown are the potential downstream effects due to interactions between mSOD1, M1 receptors and glutamate excitotoxicity.
As requested, the revised manuscript has been modified to more deeply address such issues.
The manuscript has been also carefully revised according to all the detailed reviewer’s requirements.
Reviewer 2 Report
The authors present a review examining the role of SOD1 in ALS and wild type. Data is presented from experimental models with human context. The authors have a well organized and written review. A few suggestions for improvement include:
1. Presently the review presents no organized quantitative data, metadata or aggregated analysis. I understand the objective here is not a true meta-analysis. Nonetheless, even metadata in table form would be enhance the value of the study. Even if there is no aggregation of data for each outcome, just having a synthesized lists of likely effect sizes for each of the major facets discussed would add value.
2. The authors do a good job of describing specific biology at the receptor level. However, a section on more holistic hypotheses for clinical ALS and how the receptor biology could contribute to each hypothesis would provide translational value.
3. The authors' review organization aligns pretty well with a recent large-scale 2023 meta-analysis on SOD1-G93A mice that examined the various aggregate effect sizes of pathophysiological categories on survival (including oxidative stress, inflammation, etc.) that the authors may wish to refer: https://www.frontiersin.org/articles/10.3389/fnins.2022.1111763/full
Author Response
The authors present a review examining the role of SOD1 in ALS and wild type. Data is presented from experimental models with human context. The authors have a well-organized and written review. A few suggestions for improvement include:
- Presently the review presents no organized quantitative data, metadata or aggregated analysis. I understand the objective here is not a true meta-analysis. Nonetheless, even metadata in table form would be enhance the value of the study. Even if there is no aggregation of data for each outcome, just having a synthesized lists of likely effect sizes for each of the major facets discussed would add value.
We thank the reviewer for his/her positive evaluation of our work.
ALS pathogenesis is characterized by a very complex scenario involving different, very heterogeneous, apparently unrelated molecular networks usually resulting in relentless progressive motoneuron degeneration. Here we propose SOD1-dependent pathways as a key unifying element, underlying the deranged neuron homeostasis control, that characterize the disease.
We agree with the reviewer about the relevance of metadata/aggregate analysis. However, the very heterogeneous biological scenario underlying SOD1 involvement in ALS pathogenesis/progression might seriously hamper any “more organised” evaluation. In this context, we choose to focus the complex interplay between SOD1, neurotoxicity and neuroinflammation maintaining the complexity/heterogeneity of the scenario, as has been described.
Moreover, according with the reviewer’s requests, the manuscript has been revised with the aim to provide a more unifying hypothesis for the involvement of deranged SOD1-dependent pathways (mediated by the wild-type as well as by the mutant SOD1 molecules) in ALS pathogenesis/progression.
- The authors do a good job of describing specific biology at the receptor level. However, a section on more holistic hypotheses for clinical ALS and how the receptor biology could contribute to each hypothesis would provide translational value.
This is a critical point. The aim of our manuscript, focusing the physiological role of SOD1 and the potential mechanisms underlying the deranged activity of the wild type enzyme as well as of the SOD1G93A mutant, is to open new insights on the complex interplay between SOD1, neurotoxicity and neuroinflammation. As requested, in the revised manuscript, this intriguing framework has been underlined, to propose the hypothesis that SOD1-dependent pathways might represent a unifying pathogenetic mechanism of the disease.
- The authors' review organization aligns pretty well with a recent large-scale 2023 meta-analysis on SOD1-G93A mice that examined the various aggregate effect sizes of pathophysiological categories on survival (including oxidative stress, inflammation, etc.) that the authors may wish to refer: https://www.frontiersin.org/articles/10.3389/fnins.2022.1111763/full
Thank you for this suggestion that really improve the general framework underlying our analysis. The paper of Albert JB Lee, et al., 2023 provides an intriguing description of the absolute best treatments for health status, in SOD1G93A mouse ALS model. The treatment outcome was observed to significantly vary with the disease progression. Indeed, targeting of oxidative stress in pre-onset, of excitability in near onset, of inflammation in post-onset and of apoptosis in the end-stage condition have been shown to associate with disease improvement.
In this context, the revised paper, according with the reviewer’s suggestions (that we really appreciate) has been modified to more deeply discuss SOD1 as a potential unifying molecular pathway, whose proper targeting might improve ALS treatment at different disease stage.
Round 2
Reviewer 1 Report
Page 3, line 116: Please change “finalized” to “aiming”.
Page 3, line 118: Please change “specifically” to “specially”.
Page 3, line 122-12: Please change “Data on SOD1G93A expressing neurons reveal in vitro a limited ability of such cells…” to “SOD1G93A expressing neurons cultured in vitro exhibit limited ability…”.
Page 4, line 124: Please change “positive for” to “containing”.
Page 4, line 142: Please change “associates with” to “relies on”.
Page 4, line 143: Please add “which is” before “ensured by”. Please change “loose” to “loss”.
Page 4, line 150: Please change “electron chain transport” to “electron transport chain”.
Page 4, line 151: Please change “redox mitochondrial” to “mitochondrial redox”.
Page 4, line 152: Please change “access” to “be present in”.
Page 6, line 214: Please change “immune responses against self-tissues” to “auto-immune responses”.
Page 6, line 215: Please change “CD4” to “CD4+CD25+”.
Page 6, line 233: Please change “modulating” to “tunning down”.
Page 6, line 240: Please change “instead, no data are available on” to “In contrast, there exist very limited understandings about”.
Page 9, line 362: Please put reference [107] before the period.
Page 9, line 364: Please change “at neuron post-synaptic level” to “in post-synaptic neurons”.
Page 10, line 397: Please change “strongly interfere with” to “have strong impact on”.
Page 10, line 398: Please delete “intracellular” after “mitochondrial Ca2+”.
Page 10, line 401: Please change “Ca2+ mitochondrial” to “mitochondrial Ca2+”.
Page 10, line 402: Please change “consequently interfering with” to “as well as the consequent impact on”.
Page 11, line 438: Please change the semicolon to a period.
The current version still needs to be edited by personnel using English as first language.